# Meaning in Life and the Acceptance of Cancer: A Systematic Review

**DOI:** 10.3390/ijerph19095547

**Published:** 2022-05-03

**Authors:** Rossella Mattea Quinto, Francesco De Vincenzo, Laura Campitiello, Marco Innamorati, Ekin Secinti, Luca Iani

**Affiliations:** 1Department of Human Sciences, European University of Rome, 00163 Rome, Italy; rossellamattea.quinto@unier.it (R.M.Q.); francesco.devincenzo@unier.it (F.D.V.); laura.campitiello96@gmail.com (L.C.); 2Department of Psychology, Indiana University-Purdue University Indianapolis, Indianapolis, IN 46202, USA; ekin.secinti@gmail.com

**Keywords:** acceptance, cancer, coping, meaning in life, oncology, sense of coherence, PRISMA, systematic review

## Abstract

Meaning in life and acceptance of cancer are critical for patients to adjust to a cancer diagnosis and to improve psychological wellbeing. Little is known about the relationship between meaning in life and the acceptance of cancer. This study provides a systematic review of the associations between meaning in life and the acceptance of cancer in cancer patients. CINAHL, MEDLINE, PsycINFO, and SCOPUS databases were searched until 15 March 2021. Studies were included if they quantitatively examined the association between meaning in life and the acceptance of cancer in adult cancer patients/survivors and if they were published in peer-reviewed journals or in books. The study quality was assessed using Joanna Briggs Institute critical appraisal tools. Of the 4907 records identified through database searches, only 3 studies quantitatively examined the associations between meaning in life and the acceptance of cancer. The total sample involved 464 women with cancer. All three studies reported positive correlations between meaning in life and the acceptance of cancer (ranging from *r* = 0.19 to *r* = 0.38), whereas meaning in life did not predict the acceptance of cancer. Overall, the meaning in life–acceptance relationship has not been sufficiently investigated, though it has relevant theoretical and clinical implications for coping with cancer. High-quality studies are needed to better understand the relationship between meaning in life and the acceptance of cancer.

## 1. Background

Many cancer patients have difficulty adjusting to their distressing circumstances (e.g., treatments, side effects, loss of functioning) and experience significant anxiety, depressive symptoms, cancer-related post-traumatic stress, and fear of cancer recurrence [1,2,3,4], although post-traumatic growth has also been reported [5]. Underlying this distress and difficulty in adjusting to cancer could be a loss of meaning in life and a struggle with the acceptance of cancer. Indeed, a diagnosis of cancer may threaten an individual’s sense of meaning in life and affect their life purpose and priorities.

The experience of cancer may violate one’s ability to believe that life is ordered and meaningful [6]. Patients often have difficulty explaining why they have to deal with pain, suffering, and, possibly, death [7]. They tend to experience fear, anger, a sense of confusion, and injustice [8,9] and struggle with the acceptance of cancer or “making peace with the disease” [10] (p. 308). When diagnosed with cancer, some patients attempt to answer questions about the meaning of this illness and their suffering [11,12], yet not all patients with cancer will inevitably engage in searching for meaning [13,14]. When patients are able to find positive meanings in their experience, they may have greater adjustment and improvements in their general wellbeing [15,16]. On the other hand, this searching for meaning can become futile and distressing [12] when individuals who search for meaning in life are not able to identify positive meanings [14].

According to Park’s Meaning Making Model, meaning in life refers to fundamental life assumptions through which individuals hierarchically order personal goals and subjective perceptions of coherence, purpose, and meaning [17,18]. Frankl [19] argued that individuals have an innate drive to find meaning and that failure to achieve meaning results in psychological distress. Thus, individuals tend to generate hypotheses and draw conclusions about why events occur and what they will imply for their future [20]. From this perspective, when cancer patients have a sense of meaning in their lives, they are able to perceive the world as comprehensible, feel involved and motivated by valued goals, and feel the significance of existence within a larger scheme of the world [21].

Park’s Meaning Making Model describes two levels of meaning: global and situational [17]. Global meaning refers to an individual’s general orienting system, including goals, feelings, global beliefs, and general concepts through which individuals interpret their world’s experiences. Situational meaning refers to “meaning in the context of a particular environmental encounter” [17] (p. 258). When the appraised situational meaning of the event is discrepant with the individual’s global meaning, people experience distress, which triggers meaning-making processes. Individuals engage in meaning-making efforts to reduce the discrepancy between appraised and global meaning and restore the perception of the world as meaningful [17]. Prior research has shown that a greater presence of positive meaning was prospectively associated with greater positive affect in long-term breast cancer survivors [22]. However, the search for meaning may not always lead to meaning being made. For example, a higher level of searching for meaning was found to be unrelated to meaning being made in patients with breast cancer [13]. Moreover, higher levels of search for meaning were associated with worse mental functioning and greater levels of negative affect in cancer patients [13,23].

When meaning-making efforts are successful, they result in “meanings made”, including the acceptance of cancer [17], which refers to “an active willingness to be present with cancer-related realities while giving up efforts to judge or control cancer-related appraisals or feelings” [24] (p. 29). A prior meta-analytic review has shown that higher levels of the acceptance of cancer are associated with lower distress, depressive symptoms, and anxiety symptoms [24]. The acceptance of cancer was also found to be positively associated with a high mental and physical quality of life as well as healthy levels of functioning in cancer patients [25,26].

Previous meta-analytic studies found distress to be negatively associated with both meaning in life [27] and the acceptance of cancer [24] in cancer patients. Theoretically, coming to terms with an illness or achieving a sense of acceptance has been considered as part of meaning being made [17,28]. Thus, finding meaning could reduce patients’ psychological distress, leading to acceptance of the cancer experience. Examining the association between meaning in life and acceptance could provide clinically relevant information, informing specifically tailored interventions for cancer patients. Therefore, the following research question was derived: what is the relationship between meaning in life and acceptance in patients with cancer?

## 2. Methods

### 2.1. Literature Search

This systematic review was conducted in accordance with the Preferred Reporting Items for Systematic Reviews and Meta-Analyses (PRISMA) Statement [29]. Three authors (L.I., R.M.Q., F.D.V.) developed the search strategy, which included an electronic search in four databases (e.g., MEDLINE, Scopus, PsycINFO, and CINAHL). The search strategy was based on a combination of search terms (e.g., Medical Subject Headings and keywords) related to the population (e.g., cancer) and the variables (e.g., meaning and acceptance of cancer; see Appendix A, for the full list of search terms for each database). Two filters (e.g., “Human Species” and “Age ≥ 19 years”) were applied. The electronic search was conducted until 15 March 2021. Records from each database were imported into Covidence software for removal of duplicates. Remaining records were exported to EndNote 20 for categorization of studies (e.g., abstract, type of publication) and then converted into Excel.

Three authors (L.C., R.M.Q., F.D.V.) independently screened records (L.C. screened 100%, R.M.Q. screened 50%, and F.D.V. screened 50% of the records). First, titles and abstracts were screened to exclude clearly ineligible studies (e.g., qualitative studies). Second, full-text articles of remaining records were examined. Third, reference lists of retrieved articles were hand-searched to identify potentially eligible studies. After each phase, discordances were resolved by discussion. When consensus was not reached, discrepancies between reviewers were resolved by a fourth reviewer (L.I.). Percent agreement was used to calculate interrater reliability. Authors were contacted when records lacked relevant information to determine eligibility and/or data extraction. The protocol of this systematic review was registered on PROSPERO (CRD42021270408).

### 2.2. Inclusion and Exclusion Criteria

Studies were included in the systematic review if they: (1) examined a sample of adult (≥18 years of age) cancer patients or survivors; (2) quantitatively investigated the association between meaning and acceptance of cancer; (3) were quantitative and observational (e.g., cross-sectional or longitudinal); (4) were published in peer-reviewed journals or in books and book chapters in any language; and (5) measured meaning in life and acceptance of cancer with valid self-report measures. Studies were excluded if they (1) were qualitative, reviews, meta-analyses, clinical or randomized-controlled trials, case studies, theses, dissertations, or conference presentations; (2) did not specifically measure meaning or acceptance scores (e.g., studies using the combined Meaning/Peace subscale of the Functional Assessment of Chronic Illness Therapy–Spiritual Well-Being Scale (FACIT–Sp) [30] or the combined Acceptance/Positive reinterpretation of the Brief-COPE Scale) [31]; (4) assessed meaning-related constructs which are conceptually distinct from meaning (e.g., post-traumatic growth); (5) enrolled patients with multiple diseases without performing subgroup analyses for cancer patients.

Meaning measures were selected a priori based on previous systematic reviews (see Appendix A) [20,27,32,33]. Potentially eligible studies investigated at least one component of the meaning-making model (e.g., meaning-making processes, appraised meaning, search for meaning, finding meaning, purpose in life) [17] through a specific meaning subscale. For example, studies were potentially eligible if they used the three-factor structure of the FACIT-Sp, which includes the meaning subscale [34,35,36]. Measures of acceptance of cancer were selected a priori based on the integrated model of acceptance (see Appendix A) [24]. In case of multiple papers from the same study, records with a higher number of participants were chosen.

### 2.3. Quality Appraisal

The Newcastle Ottawa Scale [37] for cohort studies and the Joanne Briggs Institute (JBI) critical appraisal checklists for analytical cross-sectional and case–control studies [38] were selected to assess the quality of records [39]. Two reviewers (R.M.Q., F.D.V.) independently assessed the quality of studies and resolved discrepancies by discussion or by consulting a third reviewer (L.I.).

### 2.4. Data Extraction

One reviewer (L.C.) extracted data from reviewed studies. Three reviewers (L.I., R.M.Q., F.D.V.) additionally checked the accuracy of data extraction. Data relevant for the review included details about participants (e.g., mean age, percentage of women, type of cancer, mean time since diagnosis), the assessment of meaning and acceptance of cancer, and their association (e.g., strength and statistical significance). Additional extracted data included year of publication, sample size, study design (e.g., cross-sectional, prospective, retrospective), data analysis (e.g., correlation, regression), and follow-up period for longitudinal studies.

## 3. Results

### 3.1. Search Results

Figure 1 shows the flowchart of study selection. The database search yielded 4907 results. After the removal of duplicates, 3068 titles and abstracts were screened. A total of 99 records were retained for full-text screening. There was 90–96% agreement between raters. After screening the full text of the 99 selected papers, 96 were excluded based on the inclusion criteria. There was 94–100% agreement between raters. Overall, three articles were included in this systematic review.

### 3.2. Characteristics of Included Studies

Details of included studies are shown in Table 1. Two studies had a cross-sectional design and involved women diagnosed with breast cancer who completed surgery; one study had a case–control design and examined women with breast or cervical cancer. Zhang et al. [40] examined the predictive role of different psychological and demographic variables on a patient’s acceptance of cancer. Kurowska et al. [41] examined whether patients with different levels of sense of coherence reported different scores in the acceptance of cancer. Kállay [42] investigated the relationships between meaning in life, positive affect and benefit-finding, post-traumatic growth, depression, negative affect, and coping in female cancer patients. Overall, the total sample included 464 participants with a mean age between 53.24 and 54 years. Most patients had a high school education or higher (66.1–72.0%); most of them were married (58.0–82.9%) and unemployed (42.8–54.0%) at the time of enrollment. Moreover, the time since diagnosis was mostly lower than two years. Most patients with breast cancer were treated with a mastectomy (74.3–100%).

### 3.3. Measurement Tools

Studies measured acceptance of cancer using the Acceptance subscale of the Brief-COPE (B-COPE) [43], the Acceptance of Disability Scale-Revised (ADS-R) [44], and the 8-items Acceptance of Illness Scale (AIS) [45]. Specifically, all three measures assess the individual’s level of effective coping with illness and disability. Studies assessed meaning, specifically the presence of meaning in life, using the Life Regard Index (LRI) [46] and the meaningfulness subscale from long and brief versions of the Sense of Coherence Scale (SOC) [47]. In particular, the LRI describes individuals’ beliefs that they are fulfilling a life framework or life goal that provides them with a highly valued understanding of their life [46] (p. 410). The meaningfulness subscale measures “the extent to which an individual feels that life makes sense, emotionally as well as cognitively; that at least some of the problems and demands encountered are worth an investment in energy, commitment, and engagement, and are welcome challenges rather than burdens” [48] (p. 156).

### 3.4. Associations between Meaning and Acceptance

All three studies reported positive correlations between meaning in life and the acceptance of cancer. Findings showed that patients with higher scores of meaningfulness reported significantly higher acceptance of cancer than those with lower scores of meaningfulness [41]. In addition, Zhang et al. [40] performed a regression analysis including age, marital status, type of surgery, as well as sense of coherence, coping styles, and social–relational quality subscales as predictors of acceptance of cancer and found that when other variables were included, meaningfulness did not significantly explain the acceptance of cancer.

### 3.5. Quality Assessment of Included Studies

The results of quality assessment using the JBI critical appraisal checklist for analytical cross-sectional studies are summarized in Table 2. Overall, the quality of the studies was low. All three studies did not identify confounding factors. In all studies, it was unclear whether patients were selected based on specified diagnostic criteria. In addition, Kurowska et al. [41] did not describe their exclusion/inclusion criteria.

## 4. Discussion

This systematic review aimed to synthetize the literature on the associations between meaning and acceptance in patients with cancer. We found a significant gap in the literature. Despite our rigorous methodology and persistent search efforts, we found only three studies that examined the associations between meaning in life and the acceptance of cancer. While the lack of more studies makes it difficult to provide firm conclusions, the findings yielded significant questions to guide future research.

We found that meaning in life was positively associated with the acceptance of cancer among women with breast or cervical cancer. Findings support that when cancer patients are able to make meaning of their circumstances (i.e., find meaning in life), they become more accepting of their cancer experience. This is consistent with Park’s Meaning Making Model [17], which posits a positive association between meaning-making and acceptance (an example of meaning made). In other words, distressed cancer patients may search for meaning, and as they succeed in meaning-making efforts (i.e., meaning made), they may become more accepting of their illness. Conversely, findings also suggest that patients who report low levels of meaning in life tend to have low acceptance of cancer, potentially reflecting failed attempts to reduce the discrepancy between the appraised meaning of cancer and global meaning (unsuccessful meaning-making efforts). The greater the discrepancy between appraised meaning and global meaning, the greater the distress experienced by patients. Indeed, previous meta-analytic reviews found distress to be negatively associated with both meaning in life [27] and acceptance of cancer [24]. These results provide further indirect support to our hypothesis on the meaning in life–acceptance association reflecting successful or unsuccessful meaning-making efforts.

On the other hand, results also showed that meaning in life failed to predict acceptance when considering other psychological (e.g., family intimacy and friendship) and sociodemographic (e.g., marital status) variables [40]. Thus, other patient characteristics may have more predictive power than meaning in life. It is possible that meaning in life may not be particularly important for the experience of acceptance for women with breast cancer after mastectomy. For example, these women may more likely use coping strategies that are not meaning-focused, such as focusing on a fighting spirit and planning and engaging in action towards eliminating the illness. Moreover, patients’ coping effectiveness may depend on cancer stage, treatment, and duration [49]. Notwithstanding, we cannot rule out that meaning in life may predict the acceptance of cancer in patients with other types of cancer.

### 4.1. Study Limitations

The present systematic review has several limitations. We only included articles from peer-reviewed journals to control for the quality of research and excluded the gray literature. However, papers written in any language were potentially eligible, allowing representativeness of available studies that were published in peer-reviewed journals. Finally, only studies involving women with breast or cervical cancer fulfilled the eligibility criteria, which limits the conclusions that can be drawn. Despite these limitations, our review sheds light on a possible clinically relevant yet understudied association between meaning in life and the acceptance of cancer.

### 4.2. Directions for Future Research

Our systematic review found a significant gap in the literature that studies have not examined the extent to which different types of meaning (meaning in life, meaning-making, search for meaning) are associated with acceptance. Indeed, the search for meaning and the presence of meaning [50] may have different effects on acceptance-related processes. However, these links have not been examined among cancer patients. One study with young adults found that at the trait level (i.e., a continuous attempt to make sense of life events), the search for meaning was associated with a lower presence of meaning and wellbeing, whereas at the state level (i.e., a fluctuating attempt to make meaning in response to life events), it was associated with a higher presence of meaning and wellbeing [51]. Moreover, the authors found that the presence of meaning mediated the relationship between the search for meaning and wellbeing. Future studies should investigate the dynamic interplay between different types of meaning in life (e.g., the search for and presence of meaning) and acceptance-related processes in cancer populations.

While theoretical models suggest that both meaning-making [17] and the acceptance of cancer [24] are critical for reducing distress and adjusting to cancer, the dynamics of the relationship between meaning and acceptance have not been clarified. For instance, whereas Park [17] conceived acceptance as the result of the meaning-making process, other scholars suggested acceptance to possibly be a part of the meaning-making process, leading to growth or meaning being made [24]. However, although Secinti et al. [24] proposed a different conceptualization of acceptance by citing a test of the mindfulness-to-meaning theory with cancer patients, Garland et al. [52] did not directly examine acceptance in their study. This raises the question of whether meaning in life and acceptance are relatively independent of each other, as they are both implied in the meaning-making process, leading to positive outcomes. As we cannot infer causality from our results, it is also possible that higher meaning in life is the effect of one’s greater acceptance of cancer. Moreover, Park [17] stated that a broader definition of meaning-making should consider a wide range of coping strategies, including positive reinterpretation and acceptance. Thus, an alternative way to conceptualize acceptance is as part of the meaning-making process. Future studies could disentangle the roles of meaning in life and acceptance by adopting a longitudinal design as well as accurately defining and measuring both these constructs.

Future studies are also needed to examine for which cancer patients, and under what circumstances, different types of meaning (e.g., meaning-making, meaning in life) affect the acceptance of cancer and subsequent distress outcomes. Specifically, future studies should further explore possible associations of the acceptance of cancer with sociodemographic and disease-related characteristics. For example, acceptance was found to be associated with age, gender, marital status, occupational level, and education level in several chronic diseases [40,53,54,55] and with income, site of cancer, and chemotherapy in cancer patients [56]. Indeed, as meaning-making is a dynamic process that unfolds over time [17], it is likely that concurrent associations do not capture possible relevant links between meaning in life and acceptance in different time frames. For example, breast cancer patients with high meaningfulness in the early stage of the disease might experience higher acceptance later in time.

### 4.3. Clinical Implications

Although the current results cannot be generalized to all clinical populations, they suggest that the relationship between meaning in life and acceptance of cancer has potential clinical implications for coping with cancer at different levels. At the individual level, meaning provides directions for people [57]. Many of the meanings that cancer survivors assign to their experience are associated with their subsequent psychological adjustment [28]. Indeed, when meaning-making efforts are successful, they result in meaning being made, including finding their purpose in life and accepting cancer. Thus, patients who can find meaning in cancer can also accept their illness and, in turn, have better emotional wellbeing and fewer psychological symptoms [58]. At the interpersonal level, dyadic meaning-making is best studied by extending and integrating the meaning-making framework to the case of the dyad [59]. Dyadic meaning-making is based on meaning-making efforts within couples, which aim to reduce the discrepancy between global meaning and appraised meaning. Indeed, it is important to consider how different types of patients’ meanings made affect not only patients’ affect the adjustment of those close to patients and vice versa [59].

More studies are needed before such findings can be translated into clinical interventions. These studies could be useful to demonstrate whether cancer patients who will be helped to find meaning will also accept their illness and show less maladjustment.

## 5. Conclusions

The results of this systematic review show that meaning in life is associated with one’s acceptance of cancer, although the former does not predict the latter. Our findings also show that we know very little about the relationship between meaning in life and the acceptance of cancer, even though it has important clinical and theoretical implications for coping with cancer. High-quality studies are needed to better understand the dynamics of the relationship between meaning in life and the acceptance of cancer.

## Figures and Tables

**Figure 1 ijerph-19-05547-f001:**
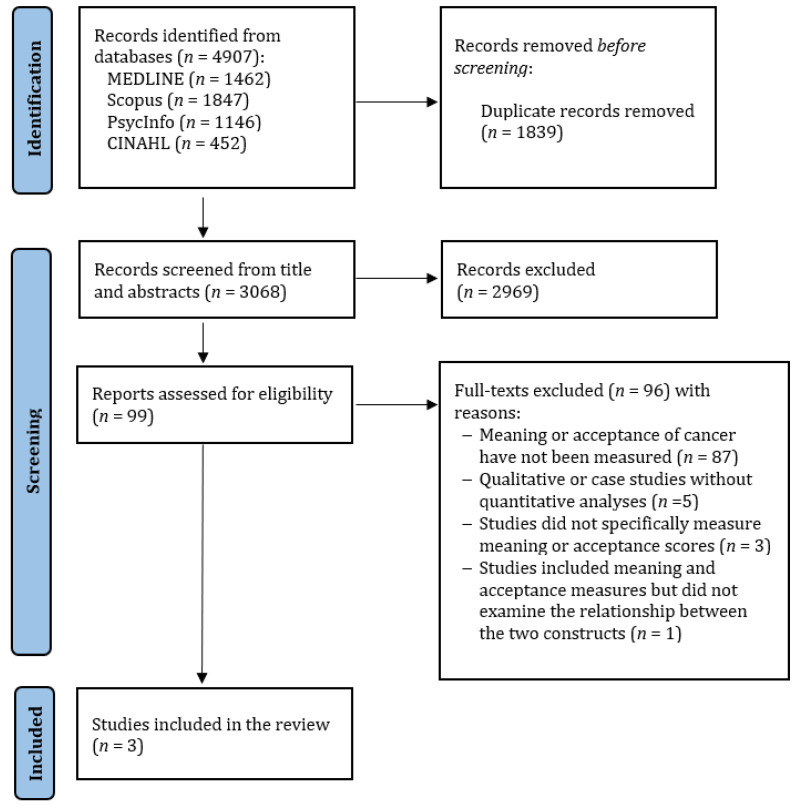
PRISMA flow diagram showing the selection of primary studies.

**Table 1 ijerph-19-05547-t001:** Studies included in the systematic review.

Study	Design	Sample	Aim	Age (Mean or Range)	Job-Employed (%)	Married (%)	Therapy Received	Disease Duration (Mean or Range)	Measures	Main Results
Kállay [42]	Case–control study	*N* = 72 Women with breast or cervical cancer	Investigate the relationship between meaning in life, positive affect and benefit-finding, post-traumatic growth, depression, negative affect, and coping in female cancer patients	53.66	NA	NA	NA	From 10 to 28 months	LRI; B-COPE	Acceptance was significantly associated with meaning in life (*r* = 0.38; *p* <0.01).
Kurowska et al. [41]	Cross-sectional study	*N* = 100 Women with breast cancer after surgery	Define the relationship between the level of coherence and illness acceptance in breast cancer patients after mastectomy	29–74	46.0	58.0	100% received mastectomy	The mean duration of cancer disease was 1.5 years	SOC-29; AIS	Illness acceptance was significantly associated with meaningfulness (*r* = 0.204). The highest scores in illness acceptance were obtained by the women who reported high levels of sense of meaning (33.36 ± 3.89)
Zhang et al. [40]	Cross-sectional study	*N* = 292 Women with breast cancer after surgery	Evaluate disability acceptance in women with breast cancer and determine the main variables associatedwith disability acceptance	53.24	57.2	82.9	25.7% received breast-conserving therapy; 74.3% received mastectomy	33.6%: less than 1 year; 38.2%: 1–2 years ago; 28.2%: 2–5 years ago.	SOC-13; AOD	Acceptance of disability was significantly and positively associated with meaningfulness (*r* = 0.196). Moreover, meaning did not predict acceptance of disability (β = 0.02; *p* = 0.576).

Note: NA = not applicable; LRI = Life Regard Index; B-COPE = Brief COPE; SOC-13 = 13-item Sense of Coherence Scale; AOD = Acceptance of Disability Scale—revised; SOC-29 = 29-item Sense of Coherence Scale; AIS = Acceptance of Illness Scale.

**Table 2 ijerph-19-05547-t002:** Critical appraisal tool according to Joanne Briggs Institute checklists.

	Q1	Q2	Q3	Q4	Q5	Q6	Q7	Q8	Q9	Q10
Analytic cross-sectional studies										
Kurowska et al. [41]	N	Y	Y	U	N	NA	Y	N	-	-
Zhang et al. [40]	Y	Y	Y	U	N	NA	Y	Y	-	-
Case-control studies										
Kállay [42]	Y	Y	U	Y	Y	N	NA	Y	Y	N

Note: Y = yes; N = no; U = unclear; NA = not applicable.

## Data Availability

The data presented in this study are available on request from the corresponding authors (L.I. and M.I.). The data are not publicly available due to privacy or ethical restrictions.

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
