# Peer review of "Meaning in Life and the Acceptance of Cancer: A Systematic Review"

_ijerph, 2022, doi:10.3390/ijerph19095547_

Round 1

Reviewer 1 Report

Thank you for the opportunity to review the manuscript: Meaning and acceptance of cancer: A systematic review.

I suggest revising the wording of the Abstract content, as is done in the MDPI International Journal of Environmental Research and Public Health.

In the Background section, the authors show evidence and the literature review is suitable for a review manuscript.

In the Methods section, the authors systematically describe the Materials and Methods used in the review.

The results of this review contribute significantly to the best available evidence for research and intervention in this field of knowledge.

The Discussion could improve, in two ways; the first to highlight the knowledge gaps that have been overcome, with the results of this systematic review, and through the available empirical evidence. And secondly, what would be the implications for the practice and intervention in three dimensions of the meaning and acceptance of cancer: individual, group and community in different populations and contexts.

Finally, it is suggested to adjust the references to the style of the MDPI International Journal of Environmental Research and Public Health.

Author Response

Thank you for the opportunity to review the manuscript: Meaning and acceptance of cancer: A systematic review.

Point 1: I suggest revising the wording of the Abstract content, as is done in the MDPI International Journal of Environmental Research and Public Health.

Response 1: We thank the reviewer for this comment. We revised the abstract according to the MPDI template (MDPI Article Template - Overleaf, Online LaTeX Editor).

In the Background section, the authors show evidence and the literature review is suitable for a review manuscript. In the Methods section, the authors systematically describe the Materials and Methods used in the review. The results of this review contribute significantly to the best available evidence for research and intervention in this field of knowledge.

Point 2: The Discussion could improve, in two ways; the first to highlight the knowledge gaps that have been overcome, with the results of this systematic review, and through the available empirical evidence. And secondly, what would be the implications for the practice and intervention in three dimensions of the meaning and acceptance of cancer: individual, group and community in different populations and contexts.

Response 2: We added the required information to the Conclusions section.

Point 3: Finally, it is suggested to adjust the references to the style of the MDPI International Journal of Environmental Research and Public Health.

Response 3: Thank you. We have changed the references according to the MDPI International Journal of Environmental Research and Public Health.

Reviewer 2 Report

Overall, it is a well written paper which outlines the current updates in understanding the relationship between meaning and acceptance of cancer. The authors have provided a comprehensive literature research with clear inclusion and exclusion criteria to highlight all the major findings of researchers in this field of study. The paper can be accepted in the present form after a few checks on grammar, spelling errors and citations.

Author Response

Overall, it is a well written paper which outlines the current updates in understanding the relationship between meaning and acceptance of cancer. The authors have provided a comprehensive literature research with clear inclusion and exclusion criteria to highlight all the major findings of researchers in this field of study.

Point 1: The paper can be accepted in the present form after a few checks on grammar, spelling errors and citations.

Response 1: Thank you. We revised the manuscript to change any spelling error.

Reviewer 3 Report

Thank You for your very important and nice work. Just a small comment:

Background: 

Overall, very informative but a bit too long. Could it be shortened?

Author Response

Thank You for your very important and nice work. Just a small comment:

Point 1: Background: 

Overall, very informative but a bit too long. Could it be shortened?

Response 1: We thank the reviewer for this suggestion. We deleted redundant information in the introduction.
